# Do Resettled People Adapt to Their Current Geographical Environment? Evidence from Poverty-Stricken Areas of Northwest Yunnan Province, China

**DOI:** 10.3390/ijerph20010193

**Published:** 2022-12-23

**Authors:** Liquan Qu, Weidong Xiao, Weidong Gao

**Affiliations:** School of Geography, Geomatics and Planning, Jiangsu Normal University, Xuzhou 221116, China

**Keywords:** poverty alleviation resettlement (PAR), adaptive capacity (AC), adaptation level (AL), differentiation mechanism, poverty-stricken areas of northwest Yunnan

## Abstract

The geographical environment adaptation of the resettled population is a deep-seated problem that determines whether the goal of the poverty alleviation resettlement (PAR) policy can be achieved. Scientific assessment of adaptive capacity (AC) and adaptation level (AL) provides a basis for subsequent policy formulation, which is of practical significance. This study took the poverty-stricken areas of northwest Yunnan as the study area and calculated the adaptive capacity index (ACI) and adaptation level index (ALI) based on survey data of 1002 resettled households and regional socioeconomic statistics by constructing the vulnerability as expected poverty (VEP) model and multi-factor analysis model. The results showed that (1) The ACI and ALI were 0.660 and 61.2 respectively, indicating that the resettled population has obvious environment adaptation barriers and a relatively high risk of returning to poverty. (2) The AC and AL of the resettled population had significant geographical differentiation. In general, Diqing Prefecture was significantly better than Nujiang Prefecture and the problems in Gongshan County, Fugong County and Lanping County were more prominent. (3) AC is a determinant of AL. However, these two indices in Gongshan and Lanping counties deviated from the general trend due to different policy effects. Based on the evaluation results and differentiation mechanism analysis, the study finally emphasized the importance of formulating and implementing the follow-up development plan of the resettled population and put forward measures to promote the resettled population to adapt to the geographical environment around the three core tasks of employment income increase, public service and bottom guarantee.

## 1. Introduction

Poverty remains an existential challenge shared by all humanity. Eradicating poverty is the primary goal of the UN sustainable development goals (SDGs) [1]. Over time, countries such as China, Brazil and Vietnam have contributed significantly to poverty alleviation around the world [2,3]. Among them, the implementation of China’s series of poverty reduction policies has lifted millions of rural poor out of poverty. Due to the complex causes of poverty [4], some areas of China were still trapped in “spatial poverty” [5]. In response, China launched a targeted poverty alleviation strategy (TPA) in 2013, which has ultimately been a great success in poverty alleviation [6]. By the end of 2020, China has eliminated the absolute poverty problem been under the current standard [7,8,9]. The central government demands that poverty alleviation areas sort out shortfalls to prevent a widespread return to poverty and view the consolidation and expansion of poverty alleviation achievements as a crucial task in the coming years.

Poverty alleviation resettlement (PAR) is an important measure of TPA [10]. PAR aims to relocate 10 million poor people living in poor conditions to suitable areas and fundamentally change their living conditions [11]. Nevertheless, resettlement under administrative intervention has not only broken the original low-level human–land symbiosis but also brought about a series of socio-economic problems such as the lack of fundamental production means, social network disruption and multi-ethnic cultural conflicts [12]. The resettled population has reached the poverty alleviation goal according to the current evaluation standard. However, they historically resided in relatively segregated areas. They are inherently sensitive and vulnerable to changes in the living environment. Therefore, we must pay attention to the resettled population. Do they adapt to the current geographical environment? What mechanisms influence their adaptation to the current geographical environment? These problems determine whether the resettled population can develop sustainably and eliminate poverty stably, which has practical significance for consolidate the successes of poverty alleviation.

This study aimed to reflect the current pattern of resettled population’s adaptation to the new environment and explain its formation mechanism by evaluating adaptive capacity (AC) and adaptation level (AL) based on a random survey of 45 relocation sites and 1002 resettled households in poverty-stricken areas of northwest Yunnan. In this way, it could provide a basis for the local formulation of measures to consolidate poverty alleviation work and reduce the risk of returning to poverty. Considering the reproducibility of China’s poverty reduction experience in developing countries, this study can also serve as a reference for relevant studies conducted in other areas of the world where carry out PAR.

## 2. Literature Review and Conceptual Framework

### 2.1. The Theory of Man–Land Areal System

The theory of man–land areal system is the cornerstone of studying the geographical environment adaptation of relocated population. According to the system theory and the theory of man–land areal system, the elements of “man” and “land” are interwoven with each other according to certain rules to form a complex and open giant system, which has a certain structure and functional mechanism inside and a certain geographical scope in space [13]. Under the interaction and mutual feedback between man and land and the external force of the system, the man–land areal system always evolves in the manner of “balance–unbalance–rebalance”, exhibiting vulnerability, risk, resilience and adaptation characteristics [14,15]. The geographical environment adaptation obstacle is the external manifestation of the mutual influence and feedback between man and nature after the sudden change of the relationship between man and land and the breaking of the system balance before and after the relocation. The core goal of man–land areal system research is to coordinate the man–land relationship [16], which is consistent with the SDGs and provides a perspective and theoretical basis for studying the geographical environment adaptation of resettled population. The basic research methods of the man–land system mainly include classification, zoning, quantitative analysis, model building and evaluation [17]. For instance, some researchers utilize the coupling degree to reflect an objective representation of the interaction stress and interaction dependence relationship between various systems or various system components and to characterize the evolution trend of the regional system at a certain moment [18,19].

### 2.2. Geographical Environment Adaptive Capacity of Migrant Population

The concept of adaptive capacity (AC) originates from natural science [20]. Steward first applied AC to human systems, studying how people adjust their behavior to the natural environment [21]. The research on the natural environment AC of population mainly involves the fields of climate change [22,23,24], disaster risk [25,26]. Due to the vulnerability of humans and the complex relationship between humans and land, there is still a lack of research on how to link daily life with adapting to environmental changes. Meanwhile, research on meso–micro scales such as cities and communities has not received much attention [27]. We retrieved all periodical literature with the keywords of “resettlement” and “adaptation” on China’s CNKI (http://www.cnki.net/, accessed on 4 May 2022) and found that the majority focuses on sociology and ethnology. It covers social adaptation [28], administrative ethical care [29], multiple spatial remodeling of migrants [30], community governance of relocated migrants [31], citizenship of relocated population [32] and social integration of relocated populations [33]. As can be observed, there is still a dearth of geographic study on the AC of the relocated population.

### 2.3. Follow-Up Development of Resettled Population

PAR aims to create conditions for subsequent poverty alleviation and development by relocating the rural poor. In developed countries, similar concepts to PAR are environmental migration [34,35] and ecological migration [36]. The research mainly involves six aspects: biodiversity conservation, poverty and pro-poor environment, climate change and regional conflicts, resettlement areas, emigration areas and sociological issues of environmental refugees. It has macro, policy, comprehensive and cross-regional characteristics [37].

Although relocation directly improves the living conditions of the resettled, the poverty problem is not resolved naturally. The poverty incidence rate of many resettlement sites is still high, so the follow-up development of the resettled population is a crucial issue. The relevant research results can be classified into two categories: one is a comprehensive study of the poverty alleviation effect and subsequent development of the resettled population [38,39]. For example, using panel data and fixed effects models, Leng et al. explored the impact of PAR on household income. They found that relocation reduces the transaction costs of poor households in accessing technology, markets and other information, improves agricultural production efficiency and the quality of agricultural products and thus leads to an increase in household farm business income [11]. Wang et al. measured the vulnerability of relocated farming households and make recommendations to increase the precision and sustainability of anti-poverty policies [40,41]. Hu et al. explored the changes in the scope of social interaction and social support under the PAR policy [42]. The second is the study of specific ways to increase income, such as the rational use of rural homesteads, employment transfer and eco-compensation [43,44,45]. These studies discuss how to improve the well-being of resettled people from different perspectives and have important reference value for relevant local governments to formulate measures to consolidate the achievements of poverty alleviation.

To sum up, the existing studies provide references for our study. However, there are still some deficiencies: first, although many scholars emphasize the importance of the follow-up development of the relocated people and put forward many measures to increase their income, few mention that the mutation of the man–land areal system and the resulting difficulty in adapting to the geographical environment are the root causes of the difficulty in the follow-up development of the resettled population and the risk of returning to poverty. Second, the current research mostly qualitatively discusses the adaptation of the resettled population to the social and cultural environment from the perspective of sociology and ethnology and the studies of quantitative evaluation of the resettled population’s geographical environment adaptation from the perspective of man–land relationship are rare. Third, China’s relocation policy applies to areas with fragile ecological environments, a lack of productive resources and difficulties in improving the living conditions of farm households, mainly in mountainous areas. Located in the Hengduan Mountains on the eastern edge of the Tibetan Plateau, Northwest Yunnan has significant geographical characteristics. First, it has a relocation population share of 14.5%, the highest in the country. Second, it has a remote national border with weak non-agricultural industries and a lack of infrastructure and public services. Third, there is a population of Tibetan, Lisu and other major ethnic minorities with low cultural quality and a tradition of living by the mountains. These factors add to the difficulties of population relocation and subsequent poverty eradication. This region has the characteristics of large-scale relocation and a high risk of returning to poverty, which should have been paid particular attention to.

Therefore, based on existing research, we took the theory of the man–land areal system as the theoretical basis and emphasized the adaptation of the population participating in PAR to the geographical environment. This study first constructed an evaluation model to quantify the AC and AL of the relocated population to the geographical environment. Secondly, using the field survey data in Northwest Yunnan, it was judged and analyzed the spatial differentiation of the AC and AL of the geographical environment of the relocated population on the county level. Through the analysis of the formation mechanism of spatial differentiation, policy recommendations to improve the AC and AL of the relocated population were finally put forward (Figure 1).

## 3. Materials and Methods

### 3.1. Study Area

The poverty-stricken areas of northwest Yunnan refer to the two ethnic minority autonomous prefectures of Diqing and Nujiang, with five counties and two cities. Nujiang Prefecture is located in the mountainous area on the border of western Yunnan, with a total population of 553,000. Ethnic minorities account for 93.9% of the population, of which the three “directly-entering-socialism ethnic groups”, Lisu, Dulong and Nu, account for 58.8%. Diqing Prefecture is located in the transition zone between Yunnan-Guizhou Plateau and Qinghai-Tibet Plateau at the junction of Yunnan, Tibet and Sichuan provinces and is adjacent to Nujiang Prefecture in the south. Of the 387,000 people in Diqing Prefecture, Tibetans, Lisu, Naxi and and other ethnic minorities account for 89.1% of the total population. Nujiang and Diqing Prefecture have similar natural environments, with mountains and rivers distributed vertically. Their reclamation coefficient is less than 4% and farmland resources are scarce. Due to the long-term constraints of unfavorable factors, this region’s socioeconomic development is relatively lagging and it is one of the poorest areas in Yunnan Province and even the whole country. Therefore, the poverty-stricken areas of northwest Yunnan have dual attributes of typicality and importance.

### 3.2. Evaluation Indicator

(1) Adaptive capacity (AC) and adaptive capacity index (ACI). AC refers to the capacity of the resettled population to cope with potential external risks and the ability to adjust themselves to deal with negative impacts [46]. It is determined by a series of household characteristic variables such as income size, demographic structure, education level of the labor force and health status of the relocating household. A relative index to it is poverty vulnerability, which expresses the likelihood that the basic living conditions of the poor will fall below the socially accepted level of the region due to household exposure to risk [47]. The number of risks that the resettled population may face and their ability to resolve risks are negatively correlated with their ability to adapt to the geographical environment. Conceptually, poverty vulnerability is an important indicator to test the adaptation of the resettled population. Therefore, we applied poverty vulnerability to measure the ACI.

Poverty vulnerability is usually defined by some relative benchmarks, such as Vulnerability as Expected Poverty (VEP) [48], Vulnerability as Low Expected Utility (VEU) [49] and Vulnerability as Uninsured Exposure to Risk (VER) [50]. Compared with VEU and VER, VEP has advantages in practical applications [51]. First, its data requirements are relatively low. Second, VEP is an ex-ante measure, which fully considers the future welfare of farmers or risks related to the future welfare and is more instructive for policy optimization [52].

VEP refers to the probability that the farmers’ expected future income is below the poverty line. Its value is subject to the distribution characteristics of the family’s future welfare level and is ultimately determined by the characteristic variables of the resettled family and the local poverty line [53]. The VEP value is between 0–1. The smaller the VEP is, the smaller the probability of future poverty, which also shows that the AC to the new environment is stronger. 1-VEP indicates the probability that the future income of farmers is higher than the poverty line, reflecting the resettled population’s environmental AC and poverty elimination ability. Therefore, this study defines 1-VEP as the environmental adaptive capacity index (ACI). ACI is also between 0 and 1. The larger the value, the stronger the AC to the new environment and vice versa.

(2) Adaptation level (AL) and adaptation level index (ALI). AL refers to the adaptation of the resettled population to the new geographical environment. It is affected by many factors such as the ability to adapt to the geographical environment, the implementation of the PAR policy and the regional natural and social environment. As PAR has a policy objective of “immigrant out of the bad environment, obtaining stability, achieving prosperity”, therefore, we measured AL by establishing an evaluation index system from these three aspects in this study.

We divided the ACI and ALI values into three intervals and then through the combination of the two, there will be nine possibilities. In addition to the “strong-high” combination, the remaining eight combinations will reflect different adaptation problems. Studying these problems and their formation mechanism is of great significance and is the basis for solving the problem.

### 3.3. Evaluation Method

#### 3.3.1. Calculation Method of ACI

(1)VEP

According to the definition of Chaudhuri et al. [53], the VEP formula is:(1)$$VEP_{i,t}=P\left(y_{i,t+1}<pl\right)$$
where $VEP_{i,t}$ represents the vulnerability of the resettled household *i* in the current period; $y_{i,t+1}$ represents the per capita income of the household in period *t +* 1, that is, the expected income; and *pl* represents the poverty line in the corresponding period; $P(y_{i,t+1}<pl)$ indicates the probability that the expected income is less than the poverty line.

It is generally believed that the income characteristics of high-income groups conform to the Pareto distribution, while the log-normal distribution is more suitable for describing the status of low-income groups [54]. For a specific period, the poverty line *pl* is the minimum annual per capita income needed to sustain a normal livelihood as measured by the level of local socioeconomic development. Therefore, let *p* = *lnpl*, Equation (1) can also be expressed as follows:(2)$$VEP_{i,t}=P\left(\ln y_{i,t+1}<p\right)$$

As mentioned above, the expected future income of the resettled farmers is mainly subject to the characteristic variables of the family, such as the education level of the main labor force, the proportion of the elderly and the health status of family members. Therefore, $\ln y_{i,t+1}$ can be expressed as a function of a set of family characteristic variables *X_i_*:(3)$$\ln y_{i,t+1}=X_{i}\beta +e_{i}$$
where *X_i_* represents the observable family characteristics of household *i*; *β* is the variable coefficient of family characteristics; $e_{i}$ is the fluctuation term of expected income. Among them, $e_{i}$ comes from the risk shock and is subject to the variable of family characteristics and obeys the following equation:(4)$$e_{i}^{2}=X_{i}\theta +\varepsilon$$
where θ is the parameter vector to be estimated; ε is the error term.

To convert the above model into an econometric model, we need to estimate the income function. This study uses the feasible generalized least square (FLGS) to estimate *β* and θ to reduce the estimation error caused by heteroscedasticity. The process is as follows:

First, the ordinary least squares (OLS) method is used to estimate Equation (3) and the estimated error term is used in Equation (4) to obtain the following:(5)$${\hat{e}}_{i}^{2}=X_{i}{\hat{\theta }}_{OLS}+\hat{\varepsilon }$$
where $\hat{\varepsilon }$ is the random error term. By dividing the both sides of Equation (4) by $X_{i}{\hat{\theta }}_{OLS}$, we can obtain the following equation:(6)$$\frac{e_{i}^{2}}{X_{i}{\hat{\theta }}_{OLS}}=\left\{\frac{X_{i}}{X_{i}{\hat{\theta }}_{OLS}}\right\}\theta +\frac{\varepsilon }{X_{i}{\hat{\theta }}_{OLS}}$$

Next, we perform OLS regression on Equation (6) to obtain the asymptotically effective estimate of θ, as follows.
(7)$${\hat{e}}_{i,FGLS}^{2}=X_{i}{\hat{\theta }}_{FGLS}\;\mathrm{or}\;{\hat{e}}_{i,FGLS}=\sqrt{X_{i}{\hat{\theta }}_{FGLS}}$$

Removing both sides of Equation (3) with in Equation (7), we obtain the following:(8)$$\frac{lny_{i,t+1}}{{\hat{e}}_{i,FGLS}}=\frac{X_{i}}{{\hat{e}}_{i,FGLS}}\beta +\frac{e_{i}}{{\hat{e}}_{i,FGLS}}$$

By estimating Equation (8), the asymptotic effective estimation value ${\hat{\beta }}_{FLGS}$ of the coefficient *β* of the family characteristic variable is obtained. According to the estimation results ${\hat{\beta }}_{FLGS}$ and ${\hat{\theta }}_{FGLS}$, we can obtain the expectation and variance of the expected logarithm of income:(9)$$\hat{E}\left(lny_{i}|X_{i}\right)=X_{i}{\hat{\beta }}_{FGLS}$$
(10)$$\hat{V}\left(lny_{i}|X_{i}\right)=X_{i}{\hat{\theta }}_{FGLS}$$

Finally, we can obtain the econometric model of VEP of resettled households:(11)$${\tilde{VEP}}_{i,t}=\tilde{P}\left(lny_{i,t+1}<p|X_{i}\right)=\emptyset \left(\frac{p-X_{i}{\hat{\beta }}_{FGLS}}{\sqrt{X_{i}{\hat{\theta }}_{FGLS}}}\right)$$

(2)ACI

According to the previous analysis of the relationship between the ACI and the VEP, the measurement model of the ACI can be expressed as follows:(12)$$\mathrm{ACI}=1-\emptyset \left(\frac{p-X_{i}{\hat{\beta }}_{FGLS}}{\sqrt{X_{i}{\hat{\theta }}_{FGLS}}}\right)$$

ACI is the probability that a relocated household’s per capita expected annual income (net of necessary expenditures) is above the poverty line. Since household characteristics variables determine expected income, constructing a regression model of the two makes it possible to quantify AC based on household characteristics variables.

(3)Family characteristic variables of resettled farmers

The expected per capita net income of the resettled households is affected by various factors such as the size of the family population, the number of laborers, the education level of the labor force, the situation of out-migrant work, the situation of receiving skill training and the situation of enjoying public welfare jobs. Since some assistance policies will be gradually canceled after the completion of poverty elimination, this study deducts rigid expenditures such as education, medical care and debt interest from the net income of households. Therefore, the family characteristic variables that affect the future expected income and rigid expenditure of the resettled households are the focus of the field survey, as detailed in Table 1.

#### 3.3.2. Calculation Method of ALI

(1)Conceptual model of AL

As mentioned above, the AL of resettled households to the geographical environment is affected by AC, degree of relocation policy implementation and other subjective and objective factors. Therefore, AL is a function of many influencing factors and the conceptual model of AL can be expressed as:(13)$$\mathrm{AL}=f\;\left(E_{1},E_{2},E_{3}\ldots \ldots E_{n}\;\right)$$
where *E_i_* represents the influencing factor; *n* represents the number of influencing factors.

(2)Index system and index weight

With the principles of comprehensiveness, weak mutual correlation, accessibility and quantification, we selected indicators of the AL to the geographic environment of the relocated population. Dimensional layer index weights were determined by the Delphi method. Indexes layer weights were determined by the hierarchical analysis method. The index weights were calculated by two comparisons using Matlab platform programming and the judgment matrix passes the consistency test and the indexes and their weights are detailed in Table 2. Finally, the weights of each index were obtained by multiplying the dimensional layer weights and the category layer weights by the multiplication method.

As seen from the table, the weights of indexes such as *X*_17_, *X*_10_, *X*_3_, *X*_19_, *X*_11_, *X*_18_, and *X*_5_ are relatively large. In the future, measures to improve the AL should be developed with a focus on these aspects.

(3)Index normalization processing

The index was dimensionless processing by the assignment method. First, we assigned a score of 100 to the best of the *j*-item indexes (*j* = 1, 2... 23) in all county-level regions. Then, according to the difference between the worst index value and the best index value, the corresponding score of the worst index value was determined, denoted as *A_jmin_* (*A_jmin_*
≥ 0). Finally, we used the range standardization Equations (14) and (15) to convert the positive and negative indicators into corresponding scores and eliminated the index units.
(14)$$A_{ij}=A_{jmin}+\frac{\left(100-A_{jmin}\right)\left(X_{ij}-X_{jmin}\right)}{X_{jmax}-X_{jmin}}$$
(15)$$A_{ij}=A_{jmin}+\frac{\left(100-A_{jmin}\right)\left(X_{jmax}-X_{ij}\right)}{X_{jmax}-X_{jmin}}$$
where *A_ij_* is the score corresponding to *j* indicators in *i* region, *X_ij_* is the index value of *j* indexes in *i* region, *X_jmax_* and *X_jmin_* are the best index value and the worst value of *j* indexes in the study area.

(4)The econometric model of ALI

According to the general method of multivariate analysis, the measurement model of ALI can be listed:(16)$$\mathrm{ALI}=\sum_{1}^{j}A_{j}W_{j}$$
where *A_j_* and *W_j_* are the scores and weights corresponding to the index *X_j_*. Because *A_j_* is a constant between 0–100 and $\sum_{1}^{j}W_{j}$ = 1, it can be seen that the ALI is between 0–100. The larger the ALI value, the higher the AL and vice versa. ALI is a relative number that reflects the relative size of the degree of adaptation to the geographical environment of the relocated population in different regions. Unlike ACI, ALI combines various factors such as household characteristics, social security and policy support and is an objective reflection of relocated households’ adaptation status to the new environment. A comparative study of ACI and ALI using counties as a unit makes it easier to identify problems in the implementation of relocation policies.

### 3.4. Data Sources

(1)Field Investigation

From August to September 2019, our research team carried out fieldwork on the geographical environment adaptation of the resettled population in the study area. In selecting the sample for the survey, we used a stratified sampling method. Firstly, relocation sites were classified according to scale, with those with more than 100 households being medium and large settlements and those with less than 100 households being small settlements. Secondly, to ensure the sample size of the survey, we randomly selected the interviewed households mainly in the medium and large resettlement sites.

The survey objects were resettlement site administrators and resettled households. The survey for resettlement site administrators is to obtain the primary conditions of the emigration and settlement areas, including the relocation distance, ethnic customs, occupancy, the resources and environment of the out-migration area and the farming methods of farmers, the industrial support of the settlement area and the construction of basic public service facilities. The survey of resettled households is mainly to grasp the family population, income source, relocation willingness, cultural inheritance, concept change, employment situation, life security and dependence on agricultural production in the place of relocation. Finally, this study investigated administrators of 45 resettlement sites and 1008 resettled households. After excluding six invalid questionnaires, we obtained 1002 valid questionnaires. The spatial distribution of the valid survey sample is shown in Figure 2.

(2)Nujiang and Diqing Prefectures Poverty Alleviation and Development Management System

The poverty alleviation and development management system registers the primary household information of all local resettled poor households and other poor households registered from the previous year of registration. It covers geographic location, family population, income breakdown, causes of poverty, poverty alleviation status, assistance policies enjoyed and effectiveness, etc. The data update every October and the results are under check and clean of the national and Yunnan provincial data, so the relevant data are more accurate and have sound currency.

(3)Statistical Bulletin of National Economic and Social Development of Diqing Prefecture and Nujiang Prefecture in 2019.

## 4. Results and Analysis

### 4.1. ACI

In the measurement of poverty elimination possibility, this study chose 15 family characteristic variables such as the proportion of population in compulsory education stage (ced), serious or long-term chronic patients (chr) as explanatory variables and chose the annual per capita income of rural households as the dependent variable to predict the future value, and calculated the probability that the annual per capita income of rural households in the next period is higher than the corresponding poverty line to judge the ability of resettled rural households to adapt to the new geographical environment. Yunnan Province used a net income per capita of CNY 4000 as the poverty line in 2020. Given that most of the resettled population lives in cities and towns and the living cost increases compared to the rural population, we used CNY 5000 as the poverty line (pl) to calculate the possibility of getting out of poverty. According to Formulas (1) to (12), we calculated the possibility of poverty elimination of 1002 resettled households in Eviews 9.0 software(HIS Global Inc., Irvine, CA, USA) and then summarized by county (city) and prefecture in the poverty-stricken areas of northwest Yunnan, as shown in Table 3.

The next step is to analyze the association between the possibility of poverty elimination of resettled population and family characteristic variables. We use the ACI as the dependent variable and the family characteristic variable as the explanatory variable and then use OLS to establish a multiple linear regression model of poverty elimination possibility:(17)$$\mathrm{AC}=\beta_{1}ced+\beta_{2}chr+\beta_{3}def+\beta_{4}edu+\beta_{5}hig+\beta_{6}lntra+\beta_{7}mla+\beta_{8}non+\beta_{9}pla+\beta_{10}sal+\beta_{11}sca+\beta_{12}wel+c$$

The results of regression analysis are shown in Table 4.

Among the explanatory variables, ced was not significant in the model, indicating that the possibility of poverty elimination has little to do with the proportion of the population in the compulsory education stage. The remaining family characteristic variables had a significant impact on AC, while non had the greatest impact on poverty elimination. Therefore, increasing income by out-migrant work is an important way to eliminate poverty. Sca, chr, def have a negative impact on AC. First, larger families often have a higher number of dependents and a smaller proportion of the labor force. This will lead to more labor constraints. Second, most families with disabled, seriously ill, or chronic patients have less labor force, higher medical expenses and lower per capita net income.

### 4.2. ALI

The AL of population to the geographical environment after relocation is affected by the AC of geographical environment, the implementation degree of relocation policy and other factors. We use the comprehensive index method to calculate the AL according to Equations (14)–(16). The calculation results are summarized by county (city) and prefecture, as shown in Table 5.

### 4.3. AC and Its Regional Differentiation Pattern

#### 4.3.1. Overall AC of Resettled Population

It can be seen from Table 4 that the geographical environment adaptive capacity of the resettled population in the poverty-stricken areas of northwest Yunnan is 0.660. This shows that the environment AC of the resettled population is weak. The probability that the expected per capita income is less than the poverty line (CNY 5000) is 34.0%. That is, there is a 34.0% risk of returning to poverty.

According to the field investigation and data analysis, the main reasons for the weak AC of the region’s resettled population are as follows. First, the income level of the resettled family is low. The per capita income of the interviewed resettled households is only CNY 8629, which is not only lower than the rural per capita disposable income level of CNY 11,902 in Yunnan Province during the same period, but also a considerable number of resettled households have income that is on the edge of the poverty line. So, their ability to resist risks is poor. Second, the income structure is unreasonable, the proportion of wage income is low and the self-development ability of resettled households is insufficient. Third, the labor force quality is generally low and inherently vulnerable to environmental changes. The average years of education of the labor force in the resettled families were 4.7 years and the proportion of “uneducated or lower than primary school” was as high as 76.2%. Among the respondents, 37.7% could understand but could not speak Mandarin and 11.6% could not understand Mandarin at all. Fourth, the long-term regional occlusion has limited communication between the resettled population and the outside world. The labor force of the interviewed families accounted for 16.9% of the local workers and even fewer went out to work, accounting for only 6.5%. 76.6% of the labor force has been engaged in traditional agricultural production for a long time and has not undergone “training to adapt to the new environment”. Fifth, the land transfer has not been kept up in time and the land operation income before the relocation has not been converted into the land property income after the relocation in time. More than 90% of the interviewed resettled households have no land transfer income and the average land transfer income accounts for only 0.2%. Sixth, the debts of some resettled households have lowered their real disposable income and weakened their AC to the environment. There are 22.5% of the interviewed households have debts. Thirty-two households have debts of CNY 50,000 to 100,000 accounting for 3.2%. Fourteen households have debts of more than CNY 100,000 accounting for 1.4% and the rest have debts below CNY 50,000.

#### 4.3.2. Spatial Differentiation Pattern and Formation Mechanism of AC

At the prefecture level, the AC of the resettled population in Nujiang Prefecture was 0.592 and that in Diqing Prefecture was 0.765, with significant regional differentiations (Figure 3). It is mainly determined by factors such as policy arrangements, regional conditions and ethnic habits. As far as policy arrangements are concerned, Diqing Prefecture is a Tibetan area and enjoys the “three districts and three prefectures” poverty alleviation policy and the traditional support policies of Tibetan areas. In comparison, Nujiang Prefecture only enjoys the “three districts and three prefectures” poverty alleviation policy. Therefore, Diqing Prefecture is better than Nujiang Prefecture in terms of infrastructure and people’s livelihood security. In addition, these two prefectures have different time schedules for poverty elimination and different levels of work progress, which also cause differences in AC. By the time of investigation, all three counties and cities in Diqing Prefecture had been lifted out of poverty. In contrast, only Gongshan County has achieved poverty elimination in Nujiang Prefecture. Lushui City, Fugong County and Lanping County plan to be lifted out of poverty by the end of 2020. In terms of regional conditions, Diqing also has a conspicuous advantage. The 214 National Road on the Yunnan-Tibet line crosses Diqing Prefecture and Shangri-La Airport has been in operation since 1999. In contrast, by 2020, the Nujiang Prefecture is still a rare “four noes” (no airports, railways, highways and inland waterways) area in China. The backward traffic causes regional blockages and the closure of people’s minds. Furthermore, thanks to the integration of tourism areas with Lijiang City, Diqing Prefecture received 22.02 million tourists in 2019 and achieved tourism revenue of CNY 26.6 billion. Part of the resettled population in Diqing Prefecture is directly involved in tourism and their AC to the environment has been improved. Although Nujiang Prefecture has good tourism resource endowments, its tourism industry is far behind. The scale of the industry is about one-fifth of that of Diqing Prefecture and the participation of ordinary people is also low. As far as ethnic habits are concerned, the Tibetans in Diqing Prefecture are nomads. They are accustomed to regional migration and have strong AC to the geographical environment of the population. However, the Lisu, Nu and Dulong peoples of Nujiang Prefecture have lived in the mountains for generations and have little communication with the outside world. They are very sensitive to the change of geographical environment and their AC to the new environment is much lower.

The AC differences to the geographical environment of the resettled population were more significant at the county level. Shangri-La City had the largest AC, followed by Deqin County, Weixi County, Lanping County, Lushui City, Fugong County and Gongshan County. The formation mechanism of county-level differences in AC was almost the same as that at the prefecture-level above, but to different degrees. For example, the degree of occlusion in Gongshan County was much more serious than in other counties and cities. Before the Dulong River Tunnel opened in 2015, Dulong River Township in Gongshan County was isolated from the outside world for half a year due to heavy snow. Before the beautiful highway project in Nujiang Prefecture was completed in October 2019, it would take 12 h to drive from Gongshan County to the prefecture for a distance of 260 km. The closure of geographical space seriously restricts the enlightenment of the population’s ideas and concepts, which is the main reason for the relatively weak ability of the local resettled population to adapt to the geographical environment.

### 4.4. AL and Its Regional Differentiation Pattern

#### 4.4.1. Overall AL of Resettled Population

The geographical environment AL of the resettled population in the poverty-stricken areas of northwest Yunnan is 61.2. There is a very tight time frame to complete the site selection and construction of over 100 resettlement sites and the identification and relocation of 135 thousand of people from 2015 to 2019. Therefore, when implementing the relocation policy, governments at all levels focus on relocation organizations, such as relocation mobilization, settlement construction, relocation compensation, old house demolition, etc. The government’s core task is to “immigrant out from the bad environment”, but it lacks systematic planning for the employment and security of the resettled population and the disposal of their original means of production and pays relatively little attention to “obtaining stability” and “achieving prosperity”. This also caused more than half of the resettled population to worry about their future life and have a low AL.

#### 4.4.2. Spatial Differentiation Pattern and Formation Mechanism of AL

The AL of the resettled population in Nujiang Prefecture (50.9) was significantly lower than that in Diqing Prefecture (78.4), as can be seen in Figure 3. This is mainly because there are gaps in the following indicators: (1) In terms of “immigrant out of the bad environment”, the completion of the relocation plan in Nujiang Prefecture is lower than that in Diqing Prefecture, while the proportion of households who have broken the contract and regretted moving out is much higher. By the time of the investigation, the relocation of Diqing Prefecture had almost ended and the contradictions and problems in the relocation had basically been effectively resolved. On the contrary, due to the late planning of poverty elimination and other special reasons in Nujiang Prefecture, its problem of “immigrant out of the bad environment” is still relatively prominent. The relocation in Nujiang Prefecture was still in progress and the relocation occupancy rate was less than 85%. Concerned about their future lives, about 6–7% of the population who have signed relocation agreements have decided to break the contract and refuse to relocate. 28.3% of the resettled households said they regretted their relocation. More than half of the resettled households indicated that they often returned to their original places of residence to facilitate agricultural production. (2) In terms of “obtaining stability”, the per capita income of resettled households in Nujiang Prefecture was CNY 7520, while that in Diqing Prefecture was CNY 10,618, which is 41.2% higher than the former. The proportion of non-agricultural income of resettled households in Diqing was 82.7%, which is also 11.2% higher than that in Nujiang. At the same time, thanks to more policy dividends, the relocation households in Diqing Prefecture are more guaranteed. The coverage rate of public welfare posts, the proportion of low-income households and the coverage rate of security coverage in Diqing Prefecture are higher than those in Nujiang Prefecture. (3) In terms of “achieving prosperity”, Diqing has better industrial support capacity and a more flexible employment pattern. In 2019, the per capita non-agricultural output value of the resident population in Diqing Prefecture was CNY 60,917.4, while that in Nujiang Prefecture was only CNY 29,954.8. The former is more than twice that of the latter. In terms of flexible employment, the number of out-migrant workers in the two prefectures is relatively small. But the proportion of the local labor force in Diqing prefecture is 25.8%, which is significantly higher than that in Nujiang prefecture, which is 12.4%.

At the county level, there are also significant differences in the actual adaptation levels of the resettled population. The ALI from high to low is Shangri-La City, Deqin County, Lushui City, Weixi County, Gongshan County, Lanping County and Fugong County. The formation mechanism of differentiations at the county level is almost the same as that at the prefecture level, except that some local factors are superimposed. For example, the Tibetan population in Deqin County is the largest. After relocation, there is almost no need for a transition period for ethnic and cultural integration and it is logically easy to adapt to the environment. There are many Christian believers in Fugong County. They are reluctant to accept government help and resist relocation, which has seriously affected the relocation progress and resulted in a low AL.

### 4.5. The Relationship between AC and AL

AC is an inherent determinant of AL. In theory, the two will show consistency, that is, if AC is weak, AL is also low. Through quantitative analysis, it was found that the linear correlation coefficient of the two indices was 0.833, indicating that there was a high positive correlation between the two indices, which was statistically significant. However, under the disturbance of external factors such as policies and national culture, the relationship between the two in some regions may also deviate from the overall trend. The AC and AL were divided into three grades with $\bar{K}\pm \delta_{K}$ as the classification mark. The AC and AL of the resettled population are consistent in Fugong County, Lushui City, Deqin County, Weixi County and Shangri-La City, while the two indicators of Gongshan County and Lanping County deviate from the overall trend: the AC of the resettled population in Gongshan County is “weak”, but the AL is “medium”, while Lanping County is the opposite, as shown in Figure 4.

The AC of the resettled population in Gongshan County is weak, but the AL is medium. This is mainly because the implementation effect of the relocation policy is better, making up for the lack of AC of the resettled population to a certain extent. Gongshan County is the only county that took the lead in alleviating poverty in the poverty elimination plan of Nujiang Prefecture. The poverty alleviation work was carried out early and vigorously. Coupled with favorable factors such as a small population and light relocation tasks, the completion of relocation work was significantly higher than that of other counties and cities in the same prefecture. At the same time, Gongshan County has implemented the PAR policy more flexibly, with fewer problems and contradictions accumulated during the relocation process. For example, according to national policy, the per capita housing of resettled households cannot exceed 20 square meters. In this way, a 40-square-meter, one-bedroom house designed for a two-person family is unsuitable for special families such as father–daughter, mother–-son, brother–sister and a house with a maximum of 140 square meters is also not suitable for a family with a particular large population. Gongshan County has effectively resolved the above-mentioned family’s reluctance to relocate by building single-person apartments in some resettlement sites. In addition, the poverty alleviation of the Dulong people has received national attention, which has brought pressure to Gongshan County and favorable policies. It has effectively promoted the poverty alleviation work of the whole county and logically improved the actual AL of the resettled population.

The situation in Lanping County is much more complicated. Lanping is located in the east of Biluoxueshan Mountain, belonging to the Dali Cultural Circle. Its lead-zinc mines are world-famous and there are a large number of immigrants, so its degree of openness is relatively much higher. The overall quality and civilized degree of the resettled population in Lanping County are relatively high and the environment AC is also strong. The reason for the low AL of the resettled population to the geographical environment is that the relocation process is slow, the relocation work has not been completed at the beginning of 2020 and various contradictions and problems accumulated in the relocation process have not been effectively resolved. Second, the relocation is large in scale and heavy in tasks. Lanping County is located in the core area of the “Three Parallel Rivers” area, with complex natural and geographical conditions. It plans to relocate 44,000 people, accounting for 26% of the county’s agricultural population and 45% of the prefecture’s relocation tasks. It is the most important resettlement task in the county of the Nujiang Prefecture. Third, more than 70% of the resettled population in the county are resettled in the urban districts. The relocation distance is long and the living environment varies greatly, which increases the uncertainty of the future life of the resettled population and inevitably increases the difficulty of relocation. Fourth, the county’s hydropower resettlement and PAR policies are not precisely connected. Some of the original hydropower resettled people included in the relocation plan refused to relocate because they thought they could not enjoy the relocation policy fairly.

## 5. Conclusions and Policy Implications

AL and AC are interrelated but distinctly different concepts, with the former indicating the objective state of adaptation and the latter reflecting the subjective adaptive capacity of relocated people to the new environment. Unlike sociological studies on the impact of individual factors such as climate, infrastructure and public services on the adaptation of the relocated population [55,56,57,58], this study provides a comprehensive evaluation and analysis of the relocated population’s adaptive capacity and adaptation level to the new environment and their geographical differentiation based on a geographical perspective and the theory of man–land relations.

Through quantitative evaluation, it was found that the AC of the resettled population to the new environment was 0.660 and the AL was 61.2 in the poverty-stricken areas of northwest Yunnan, indicating that the resettled population in this area has obvious geographical environment adaptation obstacles and a high risk of returning to poverty. The geographical environment AC of the resettled population has significant regional differences. Diqing prefecture with better regional conditions, earlier time to eliminate poverty and lighter relocation task is better than Nujiang prefecture in terms of AC and AL. At the county and city level, Shangri-La City is in the first echelon of AC, Deqin County, Lanping County, Weixi County and Lushui City are in the second echelon and Fugong County and Gongshan County are in the third echelon. The three echelons of AL are almost the same, the difference is that the positions of Lanping County and Gongshan County have been reversed.

In general, out-migrant work, public welfare jobs, the size of the family labor force, the disabled and the sick and the population above high school are the key factors affecting the environment AC of the resettled population. Long-term regional occlusion and low population quality are the deep-seated reasons for the weak AC. The AL is the result of superimposing policy effects on the basis of AC. Next, it is necessary to consolidate the achievements of poverty alleviation and prevent the return of poverty from falling into a vicious circle. It is necessary to focus on Gongshan, Fugong, Lanping and other areas, make comprehensive plans and formulate long-term measures to promote the integration of the resettled population into the new geographical environment. In the short term, the first is to encourage the labor force to go out to work through skills training, material incentives, etc. The second is to continue to improve the industrial facilities of the relocation and resettlement sites and promote the employment of some laborers nearby. The third is to maintain the existing number of public welfare posts and strengthen the management of public welfare posts to solve the employment problem of the weak labor force. The fourth is to introduce social forces while the government is leading, revitalizing the contracted land, mountain forest land and homestead resources in the resettled area and finally increasing the property income of the resettled population. As a long-term measure, it is essential to concentrate on enhancing public services in the area of relocation, as well as the quality of the service level of education, health care, pensions and minimum living security. At the same time, it insists on carrying out the project of improving the overall quality of the resettled population. The resettled population will be encouraged to change their bad habits, adjust their ideas and generally improve their quality through ongoing Mandarin training, literacy training, home life training, meaningful cultural integration and home environment evaluation activities. We must acknowledge that good environmental adaptation is challenging to achieve overnight. It takes a generation or more. For example, while the “out-migrant work” is a decisive factor in the ACI as mentioned above, the bottleneck in the growth of the migrant population in Northwest Yunnan is due to the problems of primary education and skills training, which will take a decade or even a generation to solve. Therefore, it is necessary to scientifically compile and earnestly implement the follow-up development plan of the resettled population and do a good job in the three aspects of employment income increase, public service and bottom guarantee so that the resettled population can adapt to the new geographical environment and lifestyle as soon as possible.

However, there are still some limitations to this study. Firstly, we measured the geographic adaptation of the relocated population based on household attribute data in 2019 and 2020, with a short period and little change in farm households. Although the 2019 data are field research, the focus is on relocated farm households in medium and large resettlement sites. More attention should be paid to farm households in small relocated communities. Secondly, due to COVID-19, our team was unable to collect some data in 2020 by field investigation but has to rely on the local government to update them, which may impact the accuracy of the ACI and ALI. In the post-epidemic period, optimizing the sample structure and continuously following up on research and analysis are necessary.

## Figures and Tables

**Figure 1 ijerph-20-00193-f001:**
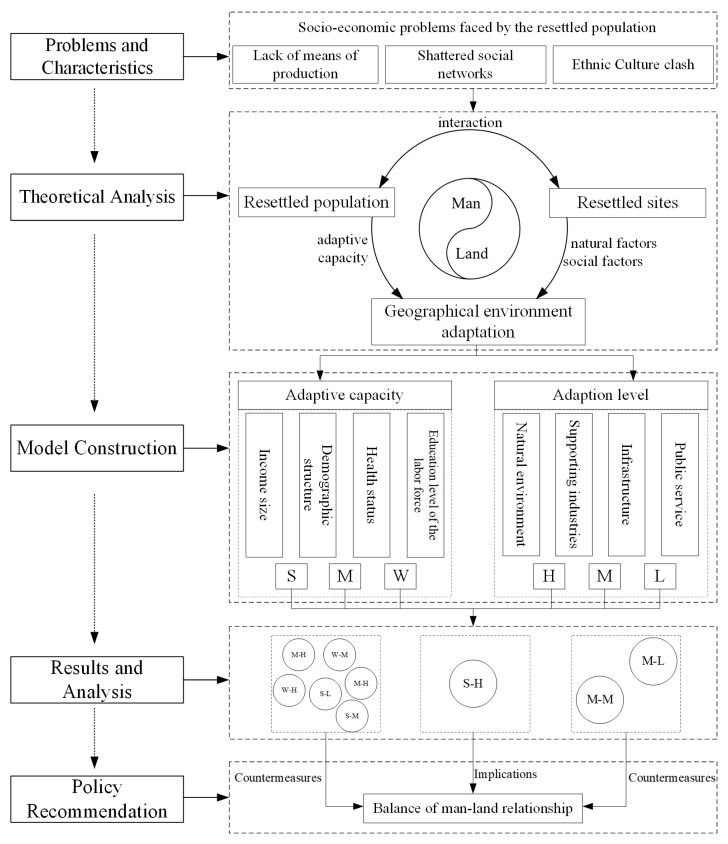
Conceptual framework. Note: In the conceptual framework, “S”, “M” and “W” represent that AC is strong, medium and weak, respectively; “H”, “M” and “L” represent that AL is high, medium and low, respectively.

**Figure 2 ijerph-20-00193-f002:**
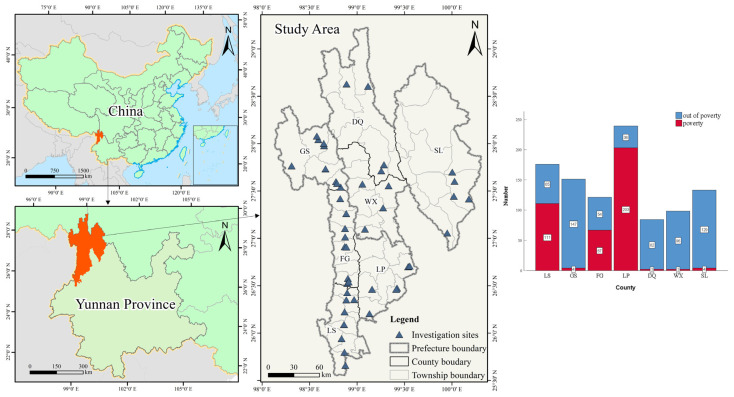
Location and basic condition of investigation sites Note: LS (Lushui City), GS (Gongshan County), FG (Fugong County), LP (Lanping County), DQ (Deqin County), WX (Weixi County), SL (Shangri-La City).

**Figure 3 ijerph-20-00193-f003:**
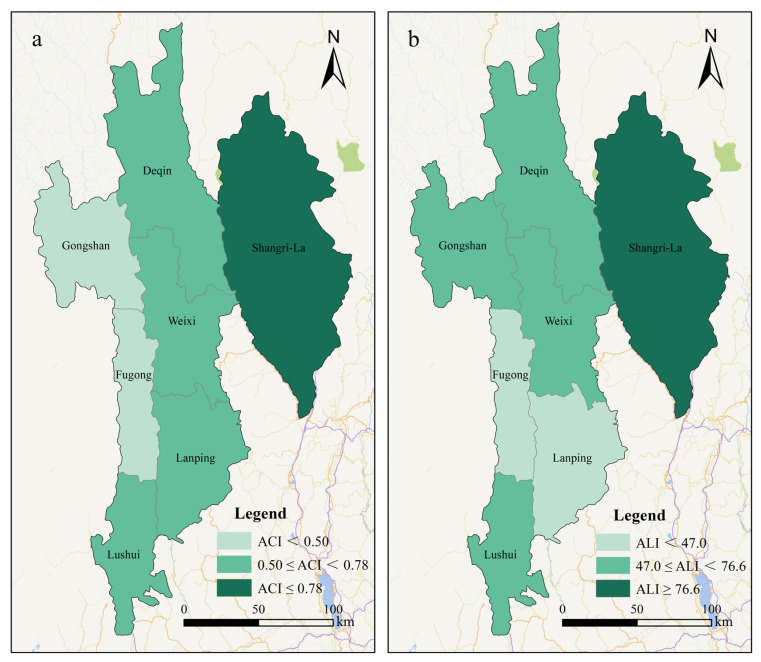
Spatial differentiation of ACI (**a**) and ALI (**b**).

**Figure 4 ijerph-20-00193-f004:**
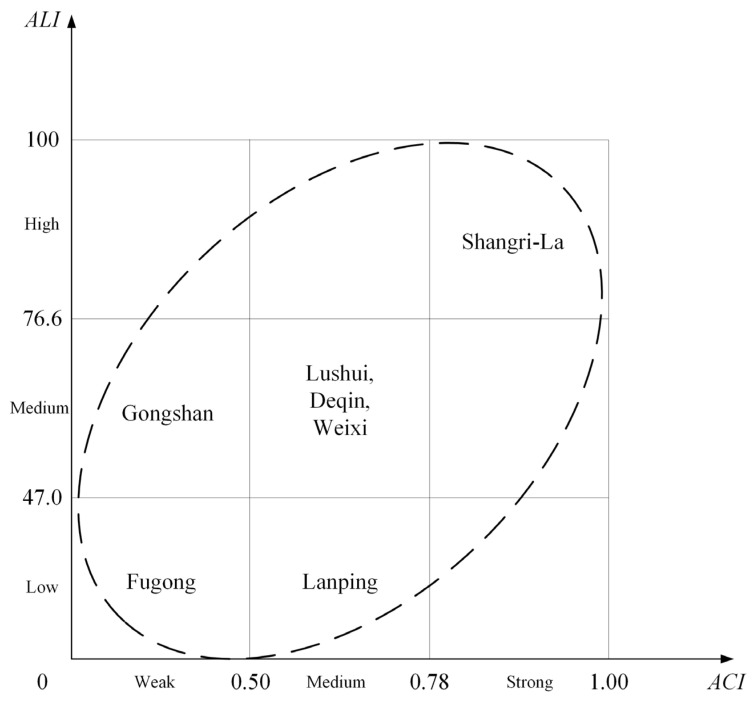
Consistency and difference of ACI and ALI. Notes: ($\bar{\mathrm{ACI}}+\delta_{\mathrm{ACI}}=0.782,\bar{\mathrm{ACI}}-\delta_{\mathrm{ACI}}=0.500;\;\bar{\mathrm{ALI}}+\delta_{\mathrm{ALI}}=76.6,\;\bar{\;\mathrm{ALI}}+\delta_{\mathrm{ALI}}=47.0$).

**Table 1 ijerph-20-00193-t001:** Household characteristic variables of resettled farmers (2020).

Variable Symbol	Variable	Description
sal	Proportion of salary income	Proportion of salary income in total household income
sca	Family scale	The total population of household registration
pla	Proportion of labor force	Labor force/total household population
non	Proportion of migrant workers population	Number of workers outside their county/total household population
chr	Serious or long-term chronic patients	Do they have serious or long-term chronic patients (the population with major diseases or long-term chronic diseases designated by rural medical insurance) at home? Yes = 1, No = 0
def	Proportion of disabled people	Number of disabled people/total family population
ced	Proportion of population in compulsory education stage	Number of people attending compulsory education/total family population
hig	Proportion of population with high school and above	Number of people attending high school, junior college, university, postgraduate/total family population
tra	Income of land transfer	Income from land transfer in the form of leases, shares, etc.
wel	Public welfare positions	Have they enjoyed the public welfare positions (including forest rangers, river managers, border guards, road guards, cleaners and so on arranged by local governments)? Yes = 1, No = 0
mla	Situation of enjoying the minimum living allowance (MLA)	Enjoy class A of MLA = 3, class B of MLA = 2, class C of MLA = 1, not = 0
edu	Education level of householder	0 for elementary school and below, 1 for middle school, 2 for high school and above

**Table 2 ijerph-20-00193-t002:** Evaluation indexes and their weights of geographical environment adaptation level.

Dimensional Layer	Category Layer	Indexes Layer		Weight
		Indexes	Explanation	
Immigrant out of the bad environment 0.20	Physical	Degree of relocation plan (*X*_1_)	Actual number of resettled households/planned number of resettled households	0.254
		Proportion of resettled households breaking the contract (*X*_2_)	Number of households refusing to relocate in breach of contract/planned number relocation households	0.170
	Psychology	Degree of relocation regret (*X*_3_)	Number of households that expressed regret after relocation/number of households resettled under field survey	0.407
		Proportion of households returning to farming (*X*_4_)	Number of households returning to farming after relocation/number of resettled households under field survey	0.169
Obtaining stability 0.45	Life and culture	Adaptation of basic living (*X*_5_)	Number of households with clean household environment and correct use of household appliances and sanitary facilities/number of resettled households	0.111
		Inheritance of ethnic culture (*X*_6_)	Good cultural heritage = 1, general = 0.5, poor = 0 (by interviewing community administrator)	0.080
		Multi-ethnic cultural integration (*X*_7_)	Good participation in cultural activities = 1, general = 0.5, poor = 0 (by interviewing community administrator)	0.096
	Social relations	Maintenance of original social relations (*X*_8_)	Number of households contacting relatives and friends “increased” and “unchanged” after relocation/number of surveyed resettled households	0.079
		Establishment of new social relations (*X*_9_)	Number of households that “communicate more and become more familiar with the resettled population that they did not know before” after relocation/number of households surveyed	0.092
	Income	Annual per capital income (*X*_10_)	Per capita net income in 2018 (unit: CNY)	0.136
		Non-agricultural income share (*X*_11_)	Household non-agricultural income/total household income	0.119
	Social security	Distance to the nearest primary school (*X*_12_)	Distance from placement to nearest primary school (unit: km)	0.052
		Distance to the nearest clinic (*X*_13_)	Distance from the resettlement site to the nearest clinic (unit: km)	0.052
		Proportion of households enjoying public welfare posts (*X*_14_)	Number of households enjoying public welfare positions/number of households resettled under investigation	0.074
		Proportion of households enjoying low insurance (*X*_15_)	Number of households enjoying low insurance/number of resettled households under investigation	0.070
		Coverage rate of bottom protection (*X*_16_)	Number of households with a bottom protection/number of households that should enjoy the bottom line	0.039
Achieving prosperity 0.35	Industry support	Per capita non-agricultural economic output value (*X*_17_)	Total output value of secondary and tertiary industries/total regional population in 2018	0.224
	Flexible employment	Proportion of local workers (*X*_18_)	Number of workers in their counties/number of labor force of resettled households under investigation	0.176
		Proportion of migrant workers (*X*_19_)	Number of migrant workers outside the county/number of labor force of resettled households under investigation	0.200
	Asset income	Per capita eco- compensation income (*X*_20_)	Eco-compensation income/household population	0.086
		Per capita land transfer income (*X*_21_)	Land transfer income/ household population	0.133
		Per capita village reclamation income (*X*_22_)	Village reclamation income/ household population	0.067
		Per capita collective economic benefits (*X*_23_)	The collective economic benefits/household population	0.114

**Table 3 ijerph-20-00193-t003:** Geographical environment ACI of the resettled population in the study area.

Regions	ACI	Regions	ACI
Nujiang Prefecture	0.592	Diqing Prefecture	0.765
Gongshan County	0.461	Shangri-La City	0.896
Fugong County	0.476	Deqin County	0.741
Lushui City	0.585	Weixi County	0.674
Lanping County	0.657	Poverty-stricken areas of northwest Yunnan	0.660

**Table 4 ijerph-20-00193-t004:** Results of multiple linear regression analysis of ACI.

**Variables**	**ced**	**chr**	**def**	**edu**	**hig**	**lntra**
Coefficient (its *t*-value)	0.0023 (0.8784)	−0.0093 * (−8.7770)	−0.0497 * (11.2477)	0.0043 * (7.5935)	0.0243 * (6.2511)	0.0079 * (39.9053)
**Variables**	**mla**	**non**	**pla**	**sal**	**sca**	**wel**
Coefficient (its *t*-value)	0.0081 * (17.8760)	0.1688 * (58.9680)	0.0670 * (34.5145)	0.0189 * (12.3062)	−0.0039 * (−30.5103)	0.0284 * (30.6524)

Note: * indicates significant at the 1% level.

**Table 5 ijerph-20-00193-t005:** Geographical environment adaptation level of resettled population in the study area.

Regions	ALI	Regions	ALI
Nujiang Prefecture	50.9	Diqing Prefecture	78.4
Gongshan County	53.1	Shangri-La City	86.5
Fugong County	43.4	Deqin County	76.0
Lushui City	65.2	Weixi County	63.8
Lanping County	44.7	Poverty-stricken areas of northwest Yunnan	61.2

## Data Availability

Not applicable.

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
