# Peer review of "Do Resettled People Adapt to Their Current Geographical Environment? Evidence from Poverty-Stricken Areas of Northwest Yunnan Province, China"

_ijerph, 2022, doi:10.3390/ijerph20010193_

Round 1

Reviewer 1 Report

The adaptive capacity index (ACI) and adaptation level index (ALI) of northwest Yunnan are calculated based on survey microdata collected and regional socioeconomic statistics using the vulnerability as expected poverty (VEP) model and multi-factor analysis model. This topic interests me greatly. Furthermore, the manuscript is well organized. I have, however, the following concerns:

1. Major issues.

(1)    In the Introduction, the third part of 1002 survey data are used in this paper. In order to ensure the scientific validity of the data, I would like to know how the author sampled the 1002 samples? It would be helpful if you could provide me with an explanation.

(2)    In 2.3. Follow-up development of resettled population, toward the end of the third paragraph, can you provide a brief explanation of why Northwest Yunnan is chosen as the research area for this paper and what distinguishes it from other deep poverty areas? It would be helpful if you could explain it in more detail. Thus, the research area selected in this paper can be made more prominent and more meaningful.

(3)    In the section on Conclusion and Policy Implications, would it be possible to include a comparison between the results of this study and those of previous similar studies? Is there a difference between the adaptation of migrants from other poor areas at the same level to their new environment and the results of this paper? As a result, the highlights of the research results in this paper can be highlighted.

(4)    In the end of Conclusion and Policy Implications, whether the authors can add some deficiencies to this paper requires further discussion in the future.

(5)    Could you tell me whether this journal has published any literature related to the research content of this paper in the past three years? Is it possible to quote it in the article?

2. Minor issues.

(1)    In Figure 1., is the word “implications” capitalized? And could you please let me know if there is a dotted line missing at the top right of Figure 1.? Would it be possible for you to check?

(2)    Is it possible to set the serial number of the coefficient in Formula 17 as a subscript?

(3)    References in this paper should be carefully checked for format. In particular, consider whether the journal names of references 13, 14, 19, 24 and 33 are consistent with those of other references. And do the references in 16 and 29 refer to Chinese journals? Would you be able to mark (in Chinese) at the end of this reference?

Reviewer 2 Report

This paper takes the deep poverty area in northwest Yunnan as the study area, and calculates the adaptive capacity index (ACI) and adaptation level index (ALI) based on survey data of 1,002 resettled households and regional  socioeconomic  statistics by establishing the vulnerability as expected  poverty  (VEP)  model  and multi-factor analysis model.

From my point of view, the topic of this paper is very interesting. The title, the keywords and the overall organization of the paper is fine.  I like this paper. I suggest this paper be accepted.

1. The explanations regarding Evaluation Method  are unclear. This is a drawback for this manuscript. Please add more explanations in Section 3.3.

2. The paper needs to include some more recent and relevant works in the area of poverty alleviation. Some examples of these works are listed below:  
-The Nexus between Credit Channels and Farm Household Vulnerability to Poverty: Evidence from Rural China
- The Reshaping of Neighboring Social Networks after Poverty Alleviation Relocation in Rural China: A Two-Year Observation
-Differences and Influencing Factors of Relative Poverty of Urban and Rural Residents in China Based on the Survey of 31 Provinces and Cities

Round 2

Reviewer 1 Report

(1)To ensure the normalization of the formula, could you edit it with Math-type?

(2)Please carefully check the format of each reference in the paper. Specially, the name of the journal that quotes each article should be an abbreviation. At the same time, the volume number should be abbreviated.
